# Response of Fibroblasts from Menkes’ and Wilson’s Copper Metabolism-Related Disorders to Ionizing Radiation: Influence of the Nucleo-Shuttling of the ATM Protein Kinase

**DOI:** 10.3390/biom13121746

**Published:** 2023-12-05

**Authors:** Laura El Nachef, Joëlle Al-Choboq, Michel Bourguignon, Nicolas Foray

**Affiliations:** 1INSERM U1296 Unit “Radiation: Defense, Health, Environment”, Centre Léon-Bérard, 69008 Lyon, France; laura.el-nachef@inserm.fr (L.E.N.); joelle.al-choboq@inserm.fr (J.A.-C.); michel.bourguignon@inserm.fr (M.B.); 2Department of Biophysics and Nuclear Medicine, Université Paris Saclay Versailles St Quentin en Yvelines, 78035 Versailles, France

**Keywords:** Menkes’ disease, Wilson’s disease, ionizing radiation, radiosensitivity, ATM protein

## Abstract

Menkes’ disease (MD) and Wilson’s disease (WD) are two major copper (Cu) metabolism-related disorders caused by mutations of the *ATP7A* and *ATP7B* ATPase gene, respectively. While Cu is involved in DNA strand breaks signaling and repair, the response of cells from both diseases to ionizing radiation, a common DNA strand breaks inducer, has not been investigated yet. To this aim, three MD and two WD skin fibroblasts lines were irradiated at two Gy X-rays and clonogenic cell survival, micronuclei, anti-*γH2AX*, -*pATM*, and -*MRE11* immunofluorescence assays were applied to evaluate the DNA double-strand breaks (DSB) recognition and repair. MD and WD cells appeared moderately radiosensitive with a delay in the radiation-induced ATM nucleo-shuttling (RIANS) associated with impairments in the DSB recognition. Such delayed RIANS was notably caused in both MD and WD cells by a highly expressed ATP7B protein that forms complexes with ATM monomers in cytoplasm. Interestingly, a Cu pre-treatment of cells may influence the activity of the MRE11 nuclease and modulate the radiobiological phenotype. Lastly, some high-passage MD cells cultured in routine may transform spontaneously becoming immortalized. Altogether, our findings suggest that exposure to ionizing radiation may impact on clinical features of MD and WD, which requires cautiousness when affected patients are submitted to radiodiagnosis and, eventually, radiotherapy.

## 1. Introduction

Copper (Cu) is a trace element essential for the metabolism of many species. In humans, Cu is the third most abundant transition metal and plays, generally as ions, a crucial role in a variety of biological processes including angiogenesis, hypoxic response, and neuromodulation. For example, Cu ions are involved in the stabilization of protein structures and the regulation of genetic expression [1,2,3]. Cu-dependent enzymes are also required in the energy metabolism (e.g., through cytochrome C oxidase), the antioxidative defense (e.g., through superoxide dismutase), and the iron metabolism (e.g., through ceruloplasmin (Cp)) [2,3]. All along the process of natural selection, cells have developed complex mechanisms to regulate intracellular Cu concentrations. Hence, some proteins may facilitate the absorption and flux of Cu ions, while others sequestrate or store them [3]. This is notably the case of Cu transporting P-type ATPases like ATP7A and ATP7B [4,5,6] (Figure 1).

In response to oxidative stress, glutathione regulates the activity of ATP7A and ATP7B ATPases via glutathionylation/deglutathionylation reactions. These two ATPases proteins incorporate Cu+ ions into the secretory pathway through the bile duct and facilitate the Cu uptake with some enzymes like Cp [3,5,7]. ATP7A and ATP7B are large, ATP-dependent, multichannel transmembrane proteins. They share about 50–60% of amino acid sequence homology [3,5] (Appendix A). Their structure consists of a cytosolic N-terminus, eight transmembrane domains (TMD) that facilitate Cu transfer via Cu translocation channel, an ATP-binding domain, and a large cytosolic C-terminus domain. The N-terminal tail contains six small metal-binding domains (MBD) capable of binding to Cu^+^ ions. MBD5 and MBD6 domains, situated close to the cellular membrane, are important for the Cu transport, while the MBD1-4 domains are involved in the catalytic activity in response to Cu exposure [3,5,6,7] (Appendix A). When Cu levels fall within the normal range, both ATP7A and ATP7B proteins have a steady-state localization at the trans-Golgi network (TGN). They facilitate the Cu-binding to Cp. An abnormally high Cu level induces a translocation of these ATPases: ATP7A navigates between the TGN and the plasma membrane, whereas ATP7B localizes between the TGN and cytosolic vesicular compartment. ATP7A is expressed in most tissues but its expression is generally low in the liver. Conversely, the ATP7B expression is abundant in the liver [3,4]. ATP7B is responsible for hepatic Cu regulation via biliary excretion by stimulating the Cu trafficking from the TGN, binding to Cp in vesicles close to the apical membrane of the hepatocytes [3,5]. Mutations of *ATP7A* and *ATP7B* genes cause Menkes’ and Wilson’s diseases, respectively [2,8].

Menkes’ disease (MD) is a rare X-linked recessive disorder that was first described by Menkes in 1962 at Columbia University in a family where five male infants showed unusual hair quality. They appeared normal at birth, but they developed later on epileptic seizures, muscular hypotonia, hypopigmentation of skin and hair, connective tissue abnormalities, and developmental disabilities leading to death caused by progressive cerebral degeneration, respiratory failure, hypothermia, and vascular complications like massive hemorrhage at the age of 7 months to 3.5 years old [8,9]. MD is caused by homozygous mutations of the *ATP7A* gene localized in Xq12-q13 [10]. More than 300 different mutations have been reported. Most of the *ATP7A* mutations were found in exons 7–10 [11,12]. The *ATP7A* mutations generally result in severe truncation and significant loss of gene function. MD incidence ranges from 1/100,000 to 1/250,000 births. Only 35–40 affected individual cases have been reported in the literature. The current treatment of MD consists of the administration of Cu-histidine, the most abundant Cu-linked amino acid found in serum. Early detection/diagnosis generally leads to a successful therapy [2,4,8,9,10]. MD is not associated with high risk of cancer but life expectancy is rather small (less than 3 years old) [2].

Wilson’s disease (WD) is a rare autosomal recessive disorder. It was initially identified by Wilson in 1912 when investigating cases showing progressive lenticular degeneration and cirrhosis of the liver [13,14]. WD is caused by homozygous or compound heterozygous *ATP7B* mutations in 13q14.3. These mutations have been associated with high levels of Cu in the brain, liver, and eyes. This Cu accumulation promotes reactive oxygen species (ROS) formation, results in a brown discoloration of the cornea called “Kayser-Fleischer ring”, liver cirrhosis, and neurological features. *ATP7B* gene was described more completely in 1993 by Bill et al. [15]. About 500 mutations have been identified [4,13,15,16,17,18]. The most frequent mutations described in Europe are His1069Glu (H1069Q) within a highly conserved domain of the ATP7B protein [19]. The average WD incidence is 1/40,000 births. The current treatment of WD is based on oral chelators like D-penicillamine (DPA) or zinc [4,13,18,20,21]. The association between WD and hepatocellular carcinoma (HCC) is still controversial: the incidence of HCC in WD patients is not statistically significant (1186 young cases of HCC has been reported in Europe according to a retrospective study) [22,23]. While iron accumulation and ROS formation in the liver are associated with an increased risk of developing HCC, most WD cases develop chronic liver inflammation and cirrhosis, but liver cancer is rare in WD patients. It is important to note that these results could be biased since all the young cases are treated with DPA, while DPA was found to prevent HCC in the Long Evans Cinnamon (LEC) rat model of WD [21,22,23].

Interestingly, significant radiosensitivity has been reported on the LEC rats both in fibroblasts [24], bone marrow stem cells [25,26], and gastrointestinal cells [27]. This last observation raises the question of the individual response of WD patients to ionizing radiation. Standard radiation therapy (RT) is not considered a good option to treat HCC in WD patients because of the sensitivity of the liver and the risks of radiation-induced cirrhosis. However, the new RT technologies like stereotactic body RT (SBRT) make possible more secured treatment thank to a more precise focus on the tumor [28]. More generally, the oxidative role of Cu ions, and the involvement of ATP7A and ATP7B ATPases in the DNA damage signaling pathways raise the question of the potential risk of radiosensitivity (radiation-induced toxicity), radiosusceptibility (radiation-induced cancer), or even radio-degeneration (radiation-induced accelerated aging) in MD and WD cells [29]. However, to our knowledge, there is still no radiobiological characterization of MD and WD cells in the literature. Hence, it appears important to us to investigate further the response of cells from MD and WD patients to ionizing radiation with or without presence of Cu.

To this aim, we proposed to apply a radiobiological routine procedure in our laboratory, which has been successfully used in a number of genetic diseases associated with radiosensitivity and/or radiosusceptibility [30,31,32,33,34,35]. This procedure is based on the radiation-induced nucleo-shuttling of the ATM protein (RIANS) and involves five different radiobiological endpoints (clonogenic cell survival, micronuclei, nuclear foci formed by H2AX, ATM, and MRE11 proteins) that were shown to be correlated together and form a very statistically robust data analysis. An X-rays dose of 2 Gy, reference-dose in radiobiology, was chosen to mimic one session of standard RT [36,37]. The RIANS procedure was applied to untransformed fibroblastic cell lines derived from three MD and two WD patients.

## 2. Materials and Methods

### 2.1. Cell Lines

To the notable exception of the study of the spontaneously transformed cells, all the experiments were performed with untransformed fibroblast cells at passages lower than 12 cultured as monolayers in the plateau phase of growth, as specified elsewhere [38]. Three radioresistant (1BR3, MRC5, Hs27) originated from radioresistant and apparently healthy individuals were used as controls. These cell lines were purchased from the European collection of authenticated cell cultures (ECACC, UK Health Security Agency, Porton Down, Salisbury, UK) under the references #90011801, #05011802, and #94041901, respectively. Two fibroblast cell lines from WD (GM05339, GM05341) and one from non-affected relative (GM05762) and three cell lines from MD (GM01057, GM00220, GM00245) and one from non-affected relative (GM04068) were purchased from the Coriell Cell Repositories (Camden, NJ, USA) (Table 1). The hyper-radiosensitive *ATM*-mutated (AT4BI) cell line was used as a positive control for radiosensitivity. This cell line belongs to the COPERNIC collection that has been abundantly documented [37]. The COPERNIC database is protected under the reference IDDN.FR.001.510017.000. D.P.2014.000.10300. All sampling protocols of the COPERNIC collection were approved by the national ethical committee in agreement with the current national regulations. The resulting cells were declared under the numbers DC2008-585, DC2011-1437, and DC2021-3957 to the Ministry of Research. Table 1 provides the major clinical and genetic features of the untransformed cell lines tested here. The SV40-transformed counterpart of the radioresistant control MRC5 cell lines, the MRC5VI (from ECACC, #85042501), served as radioresistant controls for the investigations about the spontaneously transformed MD cells. At the passages 1–12, the average doubling time of all the cell lines used in this study was 28 ± 4 h, to the notable exception of the *ATM*-mutated cells (30 ± 1 h).

### 2.2. X-rays Irradiation

Irradiations were performed with a 6 MeV photon medical irradiator (SL 15 Philips) (dose-rate: 6 Gy.min^−1^) at the anti-cancer Centre Léon-Bérard (Lyon, France) [37,38]. The dosimetry features have been certified by the Radiophysics Department of the Centre Léon-Bérard. A dose of 2 Gy has been applied to mimic one session of a standard RT [36].

### 2.3. Copper (II) Sulfate Treatment

Cells were incubated with 10 µM of Copper (II) sulfate (#451657, Sigma-Aldrich France, Quentin-Fallavier, France) diluted in ultrapure water (#10977015, Affinity, Waltham, MS, USA) and added to the culture medium for 24 h at 37 °C.

### 2.4. Clonogenic Cell Survival

The intrinsic cellular radiosensitivity was quantified from clonogenic cell survival data obtained from the standard delayed plating procedures described elsewhere [39]. The survival data were fitted to the linear-quadratic (LQ) model that describes the cell survival S as a function of dose D as follows: S = exp(−αD − βD^2^), where α and β are adjustable parameters to be determined. The surviving fraction at 2 Gy (SF2) served as a quantifiable parameter reflecting radiosensitivity [36].

### 2.5. Immunofluorescence

The immunofluorescence protocol and nuclear foci scoring were described elsewhere [32,37]. The polyclonal anti-rabbit anti-*ATP7A* antibody (#DF8506, Affinity Biosciences, Euromedex, Souffelweyersheim, France) and anti-rabbit anti-*ATP7B* antibody (#GTX30639, Genetex, Irvine, CA, USA) were used at 1:200. The anti-*γH2AX^ser139^* antibody (#05-636; Merck Millipore, Burlington, MA, USA) was used at 1:800. The monoclonal anti-mouse anti-*MRE11* (#56211) from QED Bioscience (San Diego, CA, USA) and the monoclonal anti-mouse anti-*pATM^ser1981^*(#05-740) from Merck Millipore were used at 1:100. Incubations with anti-mouse fluorescein (FITC) and rhodamine (TRITC) secondary antibodies were performed at 1:100 at 37 °C for 20 min. Slides were mounted in 4′,6′-Diamidino-2-Phenyl-indole (DAPI)-stained Vectashield (Cliniscience, Nanterre, France) and examined with an Olympus BX51 fluorescence microscope. The foci scoring procedure applied here was performed by eye and detailed elsewhere [32]. It has received the certification agreement of CE mark and ISO-13485 quality management system norms. Furthermore, the foci scoring procedure was in the frame of the Soleau Envelop and patents (FR3017625 A1, FR3045071 A1, EP3108252 A1). More than 50 nuclei were analyzed per experiment per post-irradiation time and three independent replicates were performed, at least. The Gaussian distributions of the number of foci were controlled routinely for each condition. It is noteworthy that inter-reader foci scoring has revealed no significant differences, whether it was performed by eye or by computerized ImageJ v1.5 or Olympus foci scoring software (v2.0). The size of nuclear γH2AX foci varies with post-irradiation time and their number per nucleus: the larger the number of foci, the smaller their size. Only the γH2AX foci with a size larger than 1 µm^2^ were considered [32].

### 2.6. Micronuclei Assay

During each immunofluorescence experiment, the DAPI counterstaining permits the quantification of micronuclei [40]. With such an approach, the micronuclei yield assessed may be not numerically equivalent to that obtained with the micronucleus assay involving cytochalasin B but the protocol applied facilitates the analysis of the relationship between nuclear foci and micronuclei [36,41].

### 2.7. Proximity Ligation Assay

The proximity ligation assay (PLA) allows the visualization of endogenous protein–protein interactions at the single molecule level [42]. The PLA protocol was detailed elsewhere [31,35]. The following antibodies were diluted in the Duolink antibody diluent 1X (#DUO82008, Sigma-Aldrich, Burlington, MA, USA) at a ratio of 1:100: mouse monoclonal antibody (2C1 (1A1)) anti-*ATM* (#ab78), rabbit monoclonal anti-*ATP7A* (#DF8506, Affinity, UK), and anti-rabbit anti-*ATP7B* (#GTX30639, Genetex, Irvine, CA, USA). PLA foci were assessed as the number per cell [31,35].

### 2.8. Cell Extracts and Immunoblots

The procedure of the cell extracts and immunoblots were described elsewhere [31,35]. Briefly, total extracts were obtained with the (50 mM Tris, pH 8, 150 mM NaCl, 2 mM EDTA, pH 8, 10% glycerol, 0.2% Nonidet NP40, H_2_O) lysis buffer. Cytoplasmic extracts were obtained with the (10 mM Hepes pH 7.9, 1.5 mM MgCl_2_, 10 mM KCl, 2 mM ethylenediaminetetraacetic acid (EDTA) pH 8, 0.5 mM dithiothreitol (DTT), 0.2% Nonidet NP40, H_2_O) lysis buffer [31,35]. Western blot bands were analyzed using ImageLab v6.0.1 software (Bio-Rad Laboratories, Hercules, CA, USA) [31].

### 2.9. Statistical Analysis

Two-way ANOVA was used to compare two numerical values, and Spearman’s test was used to compare the kinetic data. The foci kinetic data were fitted to the so-called Bodgi’s formula that describes the kinetics for the appearance/disappearance of nuclear foci formed by some protein relocalization after irradiation and/or genotoxic stress [43]. All the foci kinetics were done with 10 min, 1 h, 4 and 24 h post-irradiation data. Statistical analysis was performed by using Kaleidagraph v4 (Synergy Software, Reading, PA, USA) and Graphpad Prism v8 (San Diego, CA, USA).

## 3. Results

### 3.1. Cellular Radiosensitivity of MD and WD Fibroblasts

The clonogenic survival assay was applied to the fibroblasts from 3 MD (GM01057, GM00220, GM00245) and 2 WD donors (GM05339, GM05341). The resulting SF2 values were 50.6 ± 13.5%, 29.1 ± 4.3%, 58.7 ± 8.5%, 58.2 ± 8.7% and 70.0 ± 10.5%, respectively, while they were 65.5 0 ± 3.0% on average for the radioresistant controls and 2.5 ± 1% (*p* < 0.001) for the hyper-radiosensitive fibroblasts. The corresponding α and β values were: 0.3, 0.61, 0.26, 0.27, 0.17, 0.21, 1.84 Gy^−1^ for α and 0.002, 0.05, 0.001, 0.001, 0.001, 0.001, and 0.002 Gy^−2^, respectively. Only the GM00220 fibroblasts showed a radiosensitivity significantly higher than the radioresistant controls (*p* < 0.01), suggesting a strong influence of the genotype on the radiosensitivity phenotype (Figure 2). Considering the low values of the β parameters, the α/β ratios were found systematically higher than 15.

### 3.2. Abnormally High Levels of Micronuclei in MD and WD Fibroblasts

Unrepaired DNA breaks may lead to irreversibly damaged chromosomal fragments that can escape from the metaphases and generate micronuclei [29]. Micronuclei have been shown to be quantitatively correlated with cellular radiosensitivity [44]. Furthermore, fibroblasts deriving from MD patients have been shown to contain six times more Cu than controls [45]. Hence, we investigated the yields of radiation-induced micronuclei with and without a pre-treatment of 10 µM CuSO_4_ for 24 h (Figure 3).

A single pre-treatment of 10 µM CuSO_4_ for 24 h generally induces 2 to 3 residual micronuclei in human fibroblasts [44]. Without Cu pre-treatment, the number of micronuclei per 100 cells assessed 24 h post-irradiation was 5.0 ± 0.5 (GM01057) or lower in the MD cells, while it was 1.1 ± 0.5 (*p* > 0.5) and 55 ± 10 (*p* < 0.0001) in the radioresistant control and the hyper-radiosensitive fibroblasts, respectively. These data suggest that the yield of micronuclei assessed in the MD cells tested was not significantly different from the radioresistant controls. With a Cu-pretreatment, while the values for the radioresistant and hyper-radiosensitive cells did not change significantly (*p* > 0.5), the number of micronuclei assessed in the 3 MD cells tested increased significantly (*p* < 0.05 and *p* < 0.01) as indicated (Figure 3A,B).

With regard to WD cells, without Cu pre-treatment, the number of residual micronuclei per 100 cells did not exceed 2 micronuclei per 100 cells in the WD cells (*p* > 0.5). These data suggest that the number of micronuclei assessed in the WD cells tested was found similar to that of controls. Conversely, when a Cu-pretreatment was applied, the yield of micronuclei of the WD cells tested increased significantly (*p* < 0.05 or *p* < 0.01, as indicated) (Figure 3C,D). Altogether, these findings show that the Cu pre-treatment increases the yield of micronuclei in both MD and WD cells.

The mathematical link between SF2 and micronuclei has been abundantly documented in our lab. Recently, we published a relationship between these two parameters validated for 200 human fibroblasts, whatever their status of DSB recognition and repair and radiosensitivity [36]. By plotting the MD and WD data on the same graph, it appears that our data obey the published mathematical link, suggesting a strong quantitative coherence between the present study and our database (Appendix A).

### 3.3. Abnormal Number of γH2AX Foci in MD and WD Fibroblasts

The above cell survival and micronuclei data suggest that a significant subset of DNA double-strand breaks (DSB) remain unrecognized or unrepaired in MD and WD fibroblasts with or without Cu pre-treatment. To investigate the recognition and the repair of the radiation-induced DSB via the non-homologous end-joining (NHEJ) pathway, the most predominant DSB repair pathway in quiescent human cells, we applied immunofluorescence against the phosphorylated forms of the H2AX variant histone (γH2AX). In fact, in response to ionizing radiation, the phosphorylation of H2AX by the ATM protein kinase was considered as an early biomarker of the DSB recognition by the NHEJ pathway, easily detectable by the formation of nuclear foci that anti-*γH2AX* immunofluorescence reveals at the DSB sites [46] (Figure 4).

Some spontaneous γH2AX foci were observed in the MD fibroblast cell lines, but their number was not significantly different from that in the radioresistant controls (*p* > 0.5).

A single pre-treatment of 10 µM CuSO_4_ for 24 h generally induces 2 to 3 residual γH2AX foci in human fibroblasts as already published [44]. Without Cu-pretreatment, the number of γH2AX foci scored 10 min after 2 Gy was 79 ± 4 per cell, in the radioresistant controls consistent with the very documented value of 37 ± 4 per Gy per cell [47]. No γH2AX foci was observed in *ATM*-mutated cells, in agreement with literature [37]. In MD cells, the numbers of γH2AX foci scored 10 min post-irradiation were significantly lower (*p* < 0.001) than that in the radioresistant controls (GM01057, 45 ± 10, GM00220; 30 ± 10 and GM00245, 35 ± 10 γH2AX foci). At 24 h post-irradiation, the numbers of residual γH2AX foci were found similar to that of radioresistant controls (*p* > 0.1). Interestingly, the Cu pre-treatment contributed to decrease again (*p* < 0.05) the numbers of γH2AX foci scored 10 min post-irradiation (GM01057, 24.5 ± 0.5, GM00220, 25 ± 0.5 and GM00245, 25 ± 0.5 γH2AX foci), suggesting that the Cu-pretreatment may make more difficult the DSB recognition by the NHEJ pathway (Figure 4A,B).

With regard to the WD cells, again, some spontaneous γH2AX foci were observed in the WD fibroblast cell lines, but their number was not significantly different from that in the radioresistant controls (*p* > 0.5). Without Cu-pretreatment, the numbers of γH2AX foci scored 10 min post-irradiation were significantly lower (*p* < 0.001) than that in the radioresistant controls (GM05339, 53 ± 8, GM05341, 46 ± 11 γH2AX foci). At 24 h post-irradiation, the numbers of residual γH2AX foci in WD cells were found similar to that of radioresistant controls (*p* > 0.1). Like in MD cells, the Cu pretreatment contributed to decrease the numbers of γH2AX foci scored 10 min post-irradiation (GM05339, 42.5 ± 1.25, GM05341 42.5 ± 2.5 γH2AX foci) but this tendency was not found statistically significant. At 24 h post-irradiation, the numbers of residual γH2AX foci were found similar to that of non-pre-treated cells (*p* > 0.1) (Figure 4C,D).

It is noteworthy that the presence of CuSO_4_ may alter the shape of γH2AX foci since incubation with CuSO_4_ molecules was shown to induce both DNA single- (SSB) and double-strand breaks (DSB) [44]. The presence of SSB decondenses the chromatin and disperse the H2AX molecules: as a result, the size of γH2AX foci decreases and some “tiny” foci appear. This observation was also done after treatment with H_2_O_2_, an inducer of both SSB and DSB [44] or with syndromes associated with a production of SSB (e.g., [30]). The decrease of the number of γH2AX foci observed early after the pretreatment and immediately after irradiation (10 min post-irradiation) should be considered with cautiousness. For this reason, a relative error of 15% has been systematically applied to all the 10 min data [44] (see also below).

### 3.4. Abnormal Number of pATM Foci in MD and WD Fibroblasts

In the frame of the RIANS model, once in the nucleus, the radiation-induced ATM monomers reassociate after the DSB repair process to form ATM dimers, easily quantifiable as nuclear foci by immunofluorescence with the specific antibodies against the auto-phosphorylated forms of ATM (pATM) [48]. The number of pATM foci assessed after 2 Gy was 40 ± 4 in the radioresistant controls, and the hyper-radiosensitive AT4BI fibroblasts did not show any pATM foci, in agreement with previous studies [37] (Figure 5). A single pre-treatment of 10 µM CuSO_4_ for 24 h generally induces 2 to 3 residual γH2AX foci in human fibroblasts as already published [44]. In MD cells, the numbers of pATM foci scored 10 min post-irradiation were significantly lower than that in the radioresistant controls (GM01057, 25 ± 3, GM00220, 28 ± 3, GM00245, 28 ± 3 pATM foci; *p* < 0.01). At 24 h post-irradiation, the numbers of residual pATM foci were found similar to that of radioresistant controls (*p* > 0.1). Interestingly, the Cu pre-treatment contributed to decrease again (*p* < 0.05) the numbers of pATM foci scored 10 min post-irradiation (GM01057, 16 ± 2, GM00220, 16 ± 2, GM00245, 20 ± 5 pATM foci; *p* < 0.001), suggesting that the Cu-pretreatment, like with the γH2AX foci may make more difficult the DSB recognition by the NHEJ pathway (Figure 5A,B).

With regard to the WD cells, without Cu-pretreatment, the numbers of pATM foci scored 10 min post-irradiation were significantly lower (*p* < 0.001) than that in the radioresistant controls (GM05339, 30 ± 5, GM05341, 30 ± 6 pATM foci). At 24 h post-irradiation, the numbers of residual pATM foci in WD cells were found similar to that of radioresistant controls (*p* > 0.1). The Cu pretreatment contributed to maintain the numbers of pATM foci scored 10 min post-irradiation abnormally low (GM05339, 25 ± 3, GM05341, 25 ± 4 pATM foci). At 24 h post-irradiation, the numbers of residual pATM foci were found similar to that of non-pre-treated cells (*p* > 0.1).

The mathematical link between SF2 and the maximal number of pATM foci (pATMmax) has been abundantly documented in our lab. Recently, we published a relationship between these two parameters validated for 200 human fibroblasts, whatever their status of DSB recognition and repair and radiosensitivity [36]. By plotting the MD and WD data on the same relationship, it appears that our data obey the published mathematical link, suggesting a strong coherence between the present study and our database (Appendix A).

### 3.5. Abnormal Number of MRE11 Foci in MD and WD Fibroblasts

The RIANS also triggers the phosphorylation of the MRE11 nuclease protein, accompanied by the formation of nuclear MRE11 foci [49]. It is noteworthy that high numbers of MRE11 foci assessed early (first hour) post-irradiation have been observed in cells from syndromes associated with cancer proneness while high numbers of MRE11 foci assessed 24 h post-irradiation is a common feature of cells from degenerative diseases [49]. We therefore applied anti-*MRE11* immunofluorescence to the same conditions as described above. In the radioresistant controls, the MRE11 foci appeared from 2 to 8 h post-irradiation and reached their maximal yield at 4 h (7 ± 2 MRE11 foci per cell). The hyper-radiosensitive AT4BI fibroblasts did not show any MRE11 foci, in agreement with previous data [35] (Figure 6). The shape of the MRE11 foci kinetics of the MD fibroblasts appeared to be clearly different from that of the radioresistant controls with lower values of foci from 4 h post-irradiation (*p* < 0.01) (Figure 6A). After Cu-pretreatment, the shape of the MRE11 foci kinetics of MD cells changed with higher values of foci values from 2 among the 3 MD cell lines tested, whatever the time post-irradiation. Particularly, the larger number of MRE11 foci was reached at 24 h after the Cu-pretreatment while it was reached at 1 h post-irradiation without the Cu-pretreatment (Figure 6A,B). Concerning the WD cells, without the Cu-pretreatment, they showed lower values of MRE11 foci from 10 min post-irradiation (*p* < 0.01) and a maximal number of MRE11 foci reached at 24 h post-irradiation (Figure 6C). After the Cu pre-treatment of MD cells, the maximal peak was found at 1 h post-irradiation (*p* < 0.01) whereas the number of MRE11 foci was not different from zero at 24 h (Figure 6D).

### 3.6. Expression and Subcellular Localization of the ATP7A and ATP7B Proteins

A delayed RIANS is generally caused by the interaction between ATM monomers and some ATM substrates that are overexpressed in the cytoplasm [48]. Interestingly, the ATP7A protein holds 1 SQ and 1 TQ domains and the ATP7B protein holds 5 SQ and 1 TQ domains, suggesting that both proteins can theoretically interact with and be phosphorylated by ATM. We therefore examined the expression (Figure 7) and the existence of potential cytoplasmic ATM–ATP7A and ATM–ATP7B complexes (Figure 8) in MD and WD fibroblasts.

Immunoblots with total extracts indicated a very low expression of ATP7A protein (if any) in the (*ATP7A*-mutated) MD cells tested while the control Hs27 and the parental MD GM04068 cell lines elicited a significant amount of the ATP7A protein (Figure 7A). Interestingly, the ATP7B protein was found abundantly present in total extracts in MD cells although the control cell line showed the highest expression. The abundancy of the ATP7B protein was also observed in cytoplasm of the MD cells while ATP7B levels remained very low in controls (Figure 7B). All the statements were consolidated by the analysis of the grey levels (expressed in arbitrary unit) corresponding to each condition and shown in Appendix A. Similar conclusions were reached with immunofluorescence (Appendix A).

In WD cells, the ATP7B protein was found abundant in both total and cytoplasmic extracts, suggesting that, despite the *ATP7B* mutations in WD cells, some cytoplasmic ATP7B forms may be more numerous in the WD cell lines tested than in the controls or in the parental WD GM05762 cell lines (Figure 7C). Conversely, the expression of the ATP7A protein in cytoplasmic extracts of the WD cells was found low (Figure 7D). Again, grey levels analysis (Appendix A) and immunofluorescence data consolidated these conclusions (Appendix A).

As a second step, the proximity ligation assay (PLA) was applied to the MD and WD cells (Figure 8). In the MD cells, the existence of ATM–ATP7A complexes was not investigated because of the absence of the ATP7A protein supported by immunoblots (Figure 7) and immunofluorescence data (Appendix A). Conversely, we investigated the presence of ATM–ATP7B complexes in MD fibroblasts. While the number of spontaneous cytoplasmic ATM–ATP7B complexes was not found different in the MD cells tested than in controls (Figure 8A), both proteins interact much more frequently in MD cells than in controls early (10 min) after irradiation (the number of ATM–ATP7B complexes was found multiplied by 5 to 10) (Figure 8B). It is noteworthy that Cu pre-treatment did not change the conclusions significantly.

In WD cells, PLA data with anti-*ATP7A* antibodies revealed a significant but low amount of spontaneous ATM–ATP7A complexes in agreement with the low abundancy of ATP7A protein in WD cells (Appendix A). However, the number of the ATM–ATP7A complexes appeared significantly lower (*p* < 0.01) than that of ATM–ATP7B complexes observed in the same cells (Figure 8A and Appendix A). Surprisingly, in the particular case of WD cells, the exposure to irradiation did not increase the number of ATM–ATP7B complexes (Figure 8D). Conversely, PLA data were not found dependent on the Cu-pretreatment.

Altogether, our findings suggest that the ATM protein may interact in cytoplasm with ATP7B in MD cells and with ATP7A and ATP7B in WD cells. While the Cu-pre-treatment did not change such conclusions, irradiation may result in increasing the number of complexes to an extent that depends on the cell lines tested (Figure 8).

### 3.7. Caracterisation of the Transformed MD and WD Cells

During the cell culture performed in routine, no significant change of shape has been noticed in cells at the passages 1–12. By contrast, we observed a spontaneous transformation of the shape of the MD cells from passage 28–31: from a current fusiform shape of fibroblasts MD cells showed a triangle-like shape specific to epithelial cells. Such transformation was made by individual clones and their proliferation rate became drastically higher than the original fibroblast cells (Figure 9). At the same passages, the WD cells maintained their original shape (Appendix A).

In order to better understand the radiobiological consequences of such transformed MD fibroblasts isolated from clones, called GM00220T and GM00245T, we applied micronuclei, anti-*γH2AX*, -*pATM* and *MRE11*-immunofluorescence assays in the same conditions as described above (Figure 10).

In these experiments, the SV40-transformed counterpart of the radioresistant control MRC5 cell lines, the MRC5VI, served as radioresistant controls. The yield of micronuclei assessed 24 h post-irradiation was found significantly higher in the two transformed MD cells than in controls (*p* < 0.0001) (Figure 10A). With regard to γH2AX foci, the numbers of γH2AX foci assessed 10 min post-irradiation were found lower than that of controls (*p* < 0.05). The numbers of γH2AX foci assessed 24 h post-irradiation were also found higher (*p* < 0.01), suggesting both impairment of the DSB recognition and repair in the transformed MD cells (Figure 10B). The pATM data consolidated these conclusions (Figure 10C). Finally, while the formation of MRE11 foci was impaired in the GM00245T cells, the number of MRE11 reached a maximal value at 10 min and was higher at 24 h post-irradiation (Figure 10D).

## 4. Discussion

### 4.1. Justifications of the Approach

Menkes’ disease (MD) is a fatal, X-linked defect caused by *ATP7A* mutations and that results in abnormal Cu efflux from intestinal cells and inadequate Cu delivery to other tissues [8]. Conversely, Wilson’s disease (WD) is an inherited Cu toxicosis disorder resulting in defective biliary excretion attributable to a high Cu accumulation in the liver [13] (Table 2). Impairments in Cu metabolism (whether due to an excess or a lack of Cu) and/or in ATPases activity may act in the management of oxidative stress and more particularly in the recognition and repair of DNA strand breaks. Since ionizing radiation is a DNA strand breaks inducer, to document the response of these two syndromes to ionizing radiation appeared inasmuch justified as MD and WD patients may be subjected to radiodiagnosis and eventually (for WD only) to radiotherapy: an exposure to ionizing radiation may aggravate the clinical features and reduce life span. To our knowledge, this study is one of the first examples of a radiobiological characterization of these two specific diseases.

The COPERNIC procedure with the RIANS biomarkers applied on MD and WD cells has been already used on several fibroblasts deriving from patients susceptible to cancer, accelerated aging or simply treated by radiotherapy [37]. We have investigated clonogenic cellular survival, micronuclei, γH2AX, pATM, MRE11 foci, protein expression, and formation of complexes with ATM, ATP7A, and/or ATP7B proteins. From our findings, two major features emerged in our investigations in MD and WD cells:(1)a systematic impaired DSB recognition and repair(2)a systematic delayed RIANS with cytoplasmic complexation between ATM and ATP7B (for MD cells) or between ATM and both ATP7B and ATP7A (for WD cells).

### 4.2. MD and WD: Towards the Identification of the X-Proteins?

To interpret data from some hundreds fibroblast cell lines derived from cancer patients treated by radiotherapy and exhibiting severe tissue reactions, we originally proposed a mechanistic model based on the RIANS. While the ATM protein has long been considered nuclear, there is increasing evidence that ATM is also a cytoplasmic protein [50,51,52,53]. After an exposure to ionizing radiation, the cytoplasmic trans-auto-phosphorylated ATM dimers (pATM) dissociate as ATM monomers and diffuse in nucleus. Once in the nucleus, ATM phosphorylates the H2AX histones, which triggers DSB repair and phosphorylates MRE11, which inhibits its nuclease activity (Figure 11).

To be consistent with the clinical and cellular features of radiosensitivity developed above, the RIANS model has led to the definition of three groups of individuals [48,49]:-Group I that gathers radioresistant individuals whose cells elicit a normal (rapid) RIANS. This group gathers individuals with low risk of radiation-induced cancer or aging disease. After two Gy X-rays, in the group I cells, all the DSB are recognized by non-homologous end-joining (NHEJ) pathway, the most predominant DSB repair pathway in humans. The group I may represent up to 75–85% individuals [29,48].-Group II that gathers individuals whose cells elicit a delayed RIANS caused by overexpression of cytoplasmic proteins, substrates of ATM and called X-proteins. This is the group of patients at risk of cancer or else aging diseases both associated with moderate but significant radiosensitivity. Particularly, this group gathers the cancer patients at risk of post-radiotherapy radiotoxicities (with CTCAE grade higher than two but lower than five). The group II may represent up to 10–25% individuals and the syndromes are generally caused by heterozygous mutations (that generally lead to over-expression of the mutated protein) [29,48].-Group III that gathers all the individuals suffering from hyper-radiosensitivity either caused by a gross DSB recognition defect (ex: homozygous *ATM* mutations; group IIIa) or by a gross DSB repair defect (ex: homozygous *LIG*4 mutations; group IIIb. The group III gathers the very severe pediatric syndromes associated with a risk of fatal radiotherapy (CTCAE grade 5). From the above calculations, the group III may represent less than 1% individuals [29,37,48] and is generally composed of very rare recessive disorders caused by the loss of an important function.

Although MD and WD are very rare and recessive disorders, they are apparently not associated with extreme radiosensitivity like the syndromes of group III such as ataxia telangiectasia or progeria, as suggested by the clonogenic cell survival data obtained in this study: MD and WD cells are rather radioresistant but cannot belong to the group I since γH2AX, pATM, and MRE11 data suggest impairments of DSB recognition and repair. This last series of foci data strongly suggests a delayed RIANS and therefore the existence of X-proteins that, by binding to the ATM monomers may prevent the diffusion of ATM monomers in MD and WD cells: this is the common feature of the diseases belonging to the group II.

The X-protein is generally the mutated protein that causes the disease directly. However, there are some exceptions [33]. In the particular case of MD, immunoblots, immunofluorescence and PLA data revealed that ATP7A is clearly absent in MD cells in agreement with literature data [9]. Conversely, ATP7B, that is expressed in the cytoplasm of MD cells and that holds specific putative SQ and TQ domains potentially phosphorylable by ATM may serve as X-protein for MD cells. This is particularly verified after irradiation. Further investigations are needed to verify whether other proteins than ATP7B can serve as X-protein in MD cells and whether/how ATP7B may compensate the lack of ATP7A protein activity in MD cells.

With regard to WD, the scenario is more classical in the frame of the RIANS model: despite of the *ATP7B* mutations in the WD cells tested, the ATP7B protein is substantially expressed in cytoplasm and a significant number of ATM–ATP7B complexes exist, notably after irradiation whether with or without the Cu pre-treatment.

### 4.3. Potential Influence of Cu in the Response to Ionizing Radiation of MD and WD Cells

Other aspects of the radiobiological characterization of MD and WD is the specificities related to Cu. First, the quantity of Cu contained in MD cells was found five to six times higher than normal [45]. Furthermore, the presence of Cu may stimulate the ATPase activity, notably in MD cells [7,12]. Lastly, incubation of cells with Cu was shown to influence micronuclei yield, cellular proliferation, and DSB recognition and repair [44].

In our former publication related to the influence of metallic species on the DSB recognition and repair pathways [44], it was shown that 10 µM CuSO_4_ for 24 h induce up to three micronuclei, three residual γH2AX and about six mitoses per 100 cells in radioresistant control cells. Furthermore, the presence of CuSO_4_ was shown to impair DSB recognition and repair from 30 µM but not for lower Cu concentrations in radioresistant controls. However, one cannot exclude the possibility that such impairments may be observed for lower concentrations in group II cells. Indeed, in the case of MD and WD cells, the influence of endo- or exogenous Cu may be considered. First, let us recall that the ATM-dependent phosphorylation of MRE11 inactivates its nuclease activity: hence, unlike γH2AX foci, the MRE11 foci are inactivation foci [49]. Without Cu pre-treatment, both MD and WD cells showed less MRE11 foci than controls, suggesting a significant MRE11 nuclease activity. Conversely, the Cu-pre-treatment results in increasing the number of MRE11 foci by comparison to controls (both early and late MRE11 foci in MD cells, which may be associated with cellular transformation or else aging and late MRE11 foci in WD cells which may favor aging process only). However, it must be stressed that the application of the Cu pretreatment was limited for practical reasons: it was not applied with the clonogenic cell survival assay since this technique requires too long time to get colonies and would raise the question of the application of the Cu pre-treatment all along the duration of the experiment. Finally, by comparison to two Gy X-rays delivered in few minutes, the biological effect of Cu pre-treatment is very weak. and may represent 5 to 10% of the stress induced by two Gy X-rays. For this reason, we focused on the formation of ATM–ATP7A or ATM–ATP7B complexes after irradiation rather than after Cu pre-treatment. Further investigations will be needed to better determine the actual influence of Cu in the response of MD and WD cells to genotoxic stress.

### 4.4. MD and WD: At the Edge of Cancer and Degeneration?

Both MD and WD are associated with neuronal degeneration which is generally caused by accelerated aging via accumulation of endogenous and/or exogenous DNA strand breaks. In parallel, endogenous, and/or exogenous DNA strand breaks may also increase genomic instability and lead to cancer proneness.

The DNA strand breaks signaling pathways are generally managed by ordered cascades of phosphorylation to insure DNA strand breaks recognition, repair, cell cycle checkpoint, and cellular death [54]. The ATM and ATR kinases are upstream all these cascades of phosphorylation. The ATPases may impact on these cascades. Furthermore, the nuclease activity of MRE11 that is stopped by its phosphorylation by ATM also belongs to these cascades. A delayed RIANS may favor a hyper-recombination process, as systematically observed in all the cancer syndromes or else an accelerated aging process with a slow but constant increase of MRE11 foci [49]. A high concentration of Cu in cells, whether in endogenous and/or exogenous but also X-rays irradiation may therefore disturb the proportion of DNA strand breaks, the efficiency of cell cycle checkpoint and therefore condition the initiation of cellular transformation or else aging processes. The MRE11 data are representative of such fragile balance: while the Cu pre-treatment leads to early MRE11 foci in MD cells (as observed in cancer syndromes), it also increases late MRE11 foci in WD cells as observed in aging syndromes (Table 2). In parallel, without a Cu pre-treatment, the high-passages MD cells reached spontaneous cellular transformation (Figure 10) while fibroblasts from WD, that has been associated with cancer proneness did not show any sign of cellular transformation in the same conditions (Appendix A). Further experiments are therefore needed to better document the influence of ATPase, MRE11 nuclease, Cu content in the fate of MD and WD cells.

## 5. Conclusions

Altogether, our findings suggest that the MD and WD disorders are both associated with moderate radiosensitivity with notably a delayed RIANS likely caused by the sequestration of the radiation-induced ATM monomers by highly expressed ATP7B proteins that form complexes with ATM monomers in cytoplasm. Since both diseases are very rare, the number of cell lines derived from MD and WD patients available for research is low and further experiments are needed to better document the relationship genotype-phenotype. Furthermore, since the present data were performed on cutaneous fibroblasts, further experiments will be useful on other tissue types, especially liver, to better document the specificities of the radiation response associated with MD and WD syndromes. However, the literature data (e.g., [37]) have shown that cutaneous fibroblasts are sufficiently representative to suggest that cautiousness is required when affected WD and MD patients are submitted to radiodiagnosis and, eventually, radiotherapy.

## Figures and Tables

**Figure 1 biomolecules-13-01746-f001:**
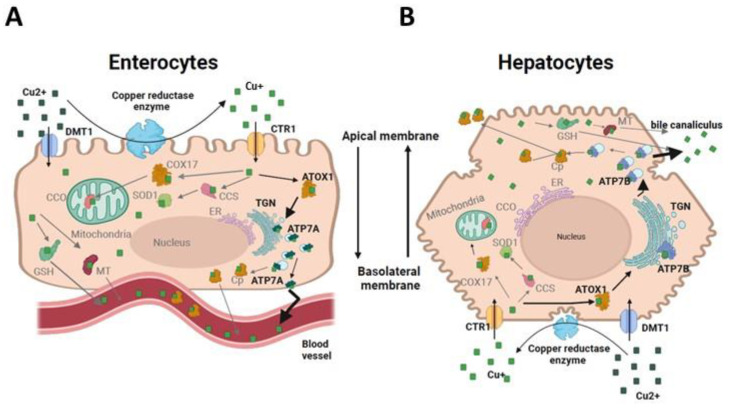
Schematic representation of the role of ATP7A and ATP7B proteins in response to Cu exposure in enterocytes (**A**) and in hepatocytes (**B**). In both cell types, Cu is initially taken up via the Cu transporter 1 (CTR1). It is then guided by the Cu chaperone ATOX1 protein to reach ATP7A or ATP7B located within the trans-Golgi network (TGN). Upon an increase in Cu levels, ATP7A and ATP7B relocate from the TGN to the outer edges of the cell. In enterocytes, ATP7A promotes Cu excretion into the bloodstream from the basolateral side, while in hepatocytes, Cu is expelled from the apical side into the bile [3,5].

**Figure 2 biomolecules-13-01746-f002:**
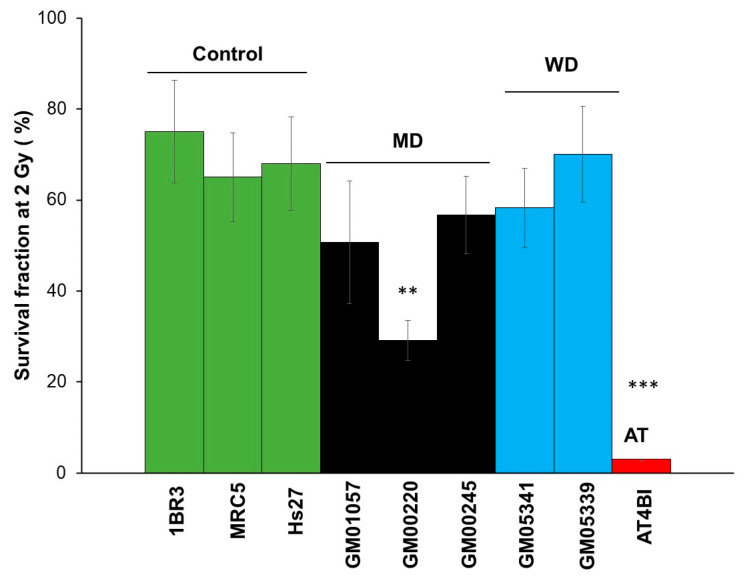
Clonogenic cell survival of MD and WD fibroblasts. A clonogenic cell survival assay was applied at 2 Gy to the radioresistant control (1BR3, MRC5, Hs27), the hyper-radiosensitive ATM-mutated (AT4BI), the MD (GM01057, GM00220 and GM00245) and the WD (GM05339, GM05341) fibroblast cell lines. Each bar represents the mean ± standard error of the mean (SEM) of three replicates. ** and *** asterisks represent a *p* < 0.01 and *p* < 0.001 difference by comparison to controls data, respectively. AT: ataxia telangiectasia.

**Figure 3 biomolecules-13-01746-f003:**
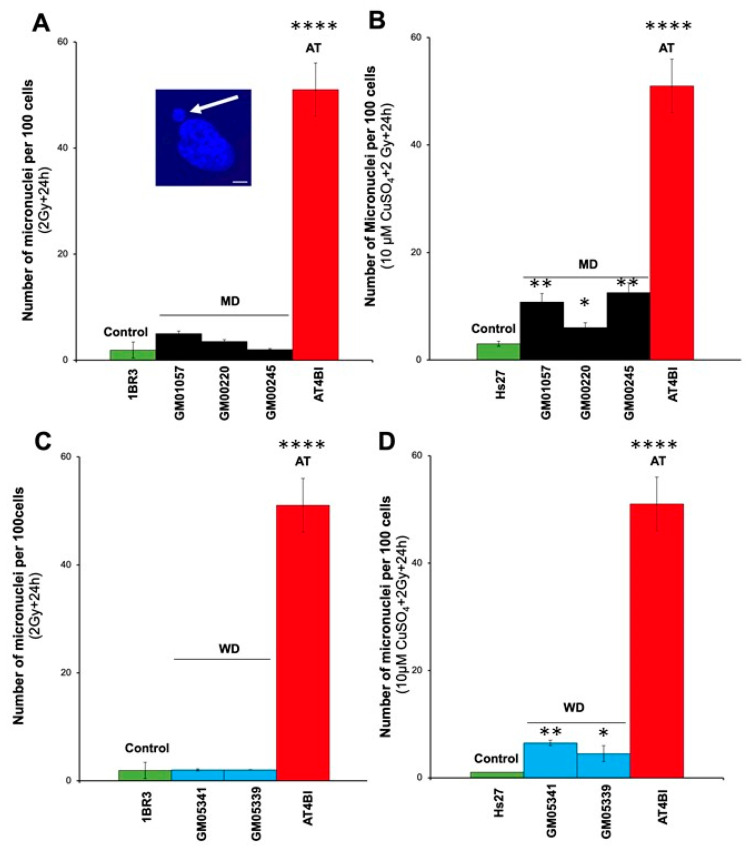
Micronuclei in MD and WD fibroblasts. Number of micronuclei per 100 cells assessed 24 h after 2 Gy X-rays in the radioresistant control (1BR3, Hs27), the hyper-radiosensitive ATM-mutated (AT4BI), the MD (GM01057, GM00220, GM00245) (**A**,**B**) and the WD (GM05339, GM05341) (**C**,**D**) fibroblast cell lines without (**A**,**C**) or with a pretreatment of 10 μM CuSO_4_ for 24 h before irradiation (**B**,**D**). Each plot represents the mean ± SEM of three replicates. The insert in (**A**) shows a representative example of a micronucleus (white arrow) observed with DAPI counterstaining. The white bar represents 30 µm. *, ** and, **** asterisks represent a *p* < 0.05, *p* < 0.01 and *p* < 0.0001 difference by comparison to controls data, respectively. AT: ataxia telangiectasia.

**Figure 4 biomolecules-13-01746-f004:**
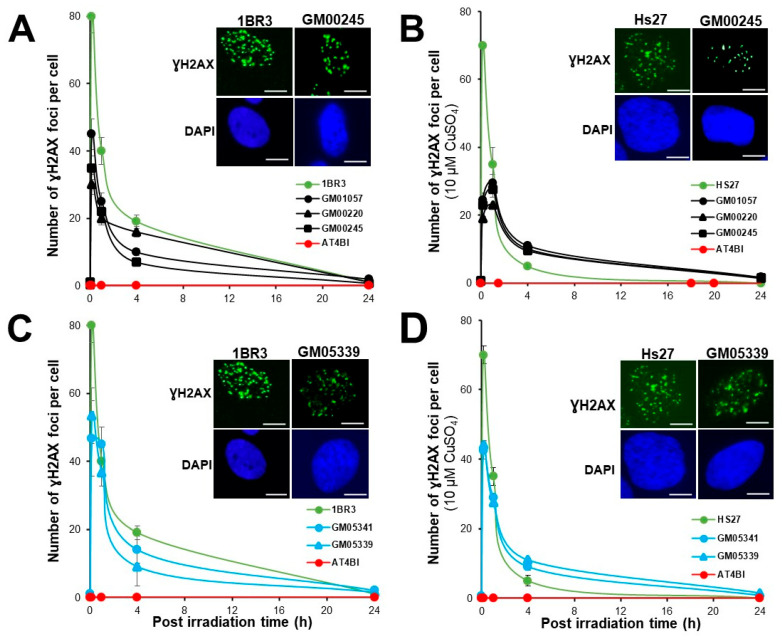
Kinetics of γH2AX foci in MD and WD fibroblasts with 10 min, 1, 4 and 24 h. Anti-γH2AX immunofluorescence was applied to the radioresistant control (1BR3, Hs27), the hyper-radiosensitive ATM-mutated (AT4BI), the MD (GM01057, GM00220, GM00245) (**A**,**B**) and the WD (GM05339, GM05341) (**C**,**D**) fibroblast cell lines without (**A**,**C**) or with a pretreatment of 10 μM CuSO_4_ for 24 h before irradiation (**B**,**D**). Each plot represents the mean ± SEM of three replicates. The inserts show representative images of γH2AX foci at 10 min post-irradiation. The white bar represents 30 µm.

**Figure 5 biomolecules-13-01746-f005:**
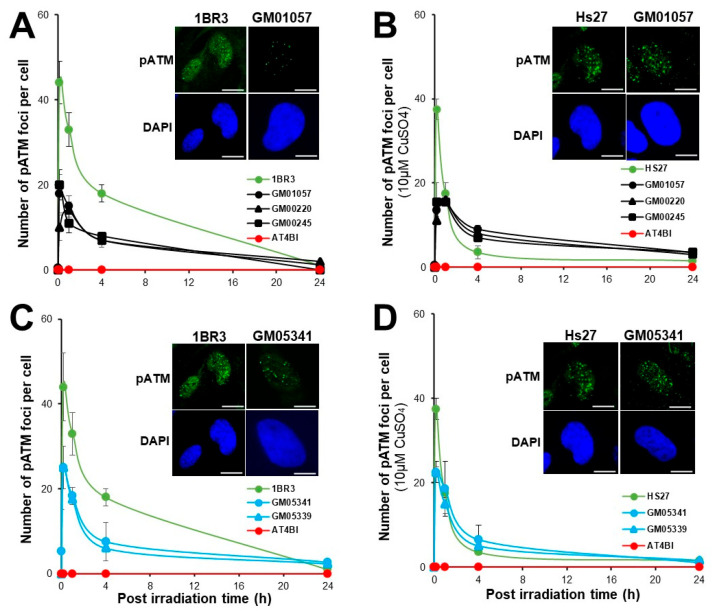
Kinetics of pATM foci in MD and WD fibroblasts with 10 min, 1, 4 and 24 h. Anti-pATM immunofluorescence was applied to the radioresistant control (1BR3, Hs27), the hyper-radiosensitive ATM-mutated (AT4BI), the MD (GM01057, GM00220, GM00245) (**A**,**B**), and the WD (GM05339, GM05341) (**C**,**D**) fibroblast cell lines without (**A**,**C**) or with a pretreatment of 10 μM CuSO_4_ for 24 h before irradiation (**B**,**D**). Each plot represents the mean ± SEM of three replicates. The inserts show representative images of pATM foci. The white bar represents 30 µm.

**Figure 6 biomolecules-13-01746-f006:**
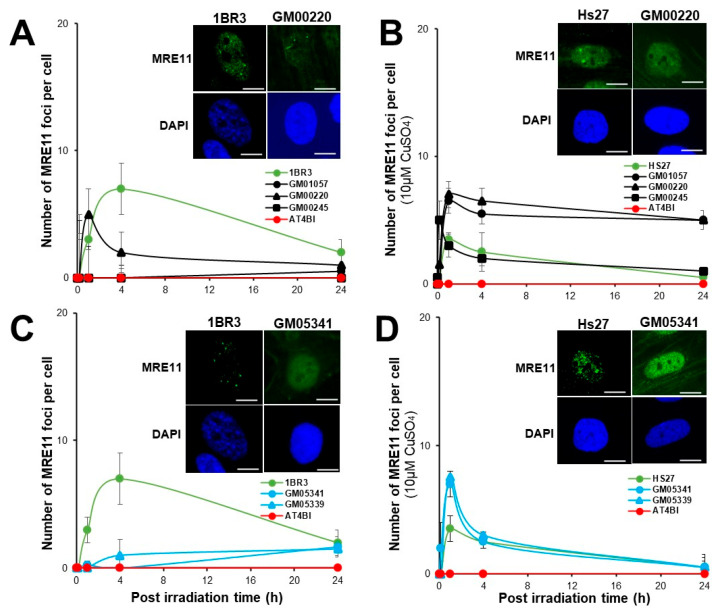
Kinetics of MRE11 foci in MD and WD fibroblasts with 10 min, 1, 4, and 24 h. Anti-MRE11 immunofluorescence was applied to the radioresistant control (1BR3, Hs27), the hyper-radiosensitive ATM-mutated (AT4BI), the MD (GM01057, GM00220, GM00245) (**A**,**B**), and the WD (GM05339, GM05341) (**C**,**D**) fibroblast cell lines without (**A**,**C**) or with a pretreatment of 10 μM CuSO_4_ for 24 h before irradiation (**B**,**D**). Each plot represents the mean ± SEM of three replicates. The inserts show representative images of MRE11 foci. The white bar represents 30 µm.

**Figure 7 biomolecules-13-01746-f007:**
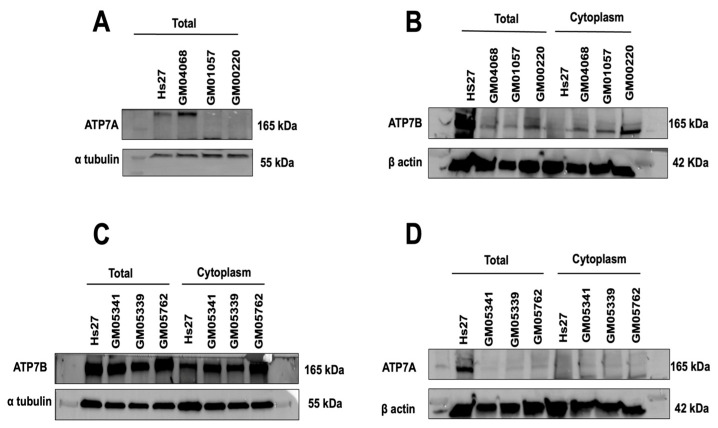
Expression of the ATP7A and ATP7B proteins in MD and WD fibroblasts. Anti-ATP7A and anti-ATP7B immunoblots with total or cytoplasmic protein extracts were applied to the indicated non-irradiated control Hs27, MD (**A**,**B**), and WD (**C**,**D**) fibroblasts. The grey levels corresponding to each condition are shown in Appendix A. Original western blot images can be found in raw data.

**Figure 8 biomolecules-13-01746-f008:**
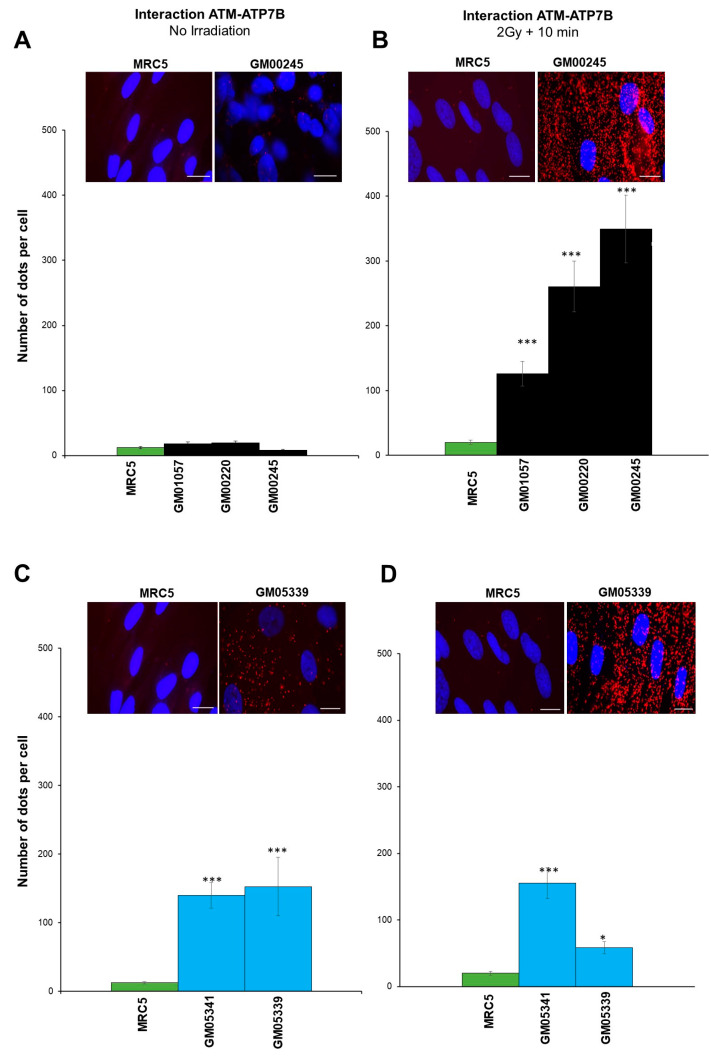
Interaction between ATM and ATP7A or ATP7B proteins observed by applying the proximity ligation assay (PLA). The PLA was applied to the indicated MD (**A**,**B**) and WD (**C**,**D**) cell lines that were exposed (**B**,**D**) or not to two Gy X-rays followed by 10 min (**A**,**C**). Representative PLA images of ATM–ATP7A or ATM–ATP7B complexes observed in the indicated MD, WD, and control cells. The nuclei were counterstained with DAPI (blue). The red foci indicate an ATM-ATP7 protein complex. The average numbers of red foci were scored per 100 cells. Each data point represents the mean ± SEM of two independent replicates. * and *** asterisks represent a *p* < 0.05 and *p* < 0.001 difference by comparison to controls data, respectively. White bars represent 30 µm.

**Figure 9 biomolecules-13-01746-f009:**
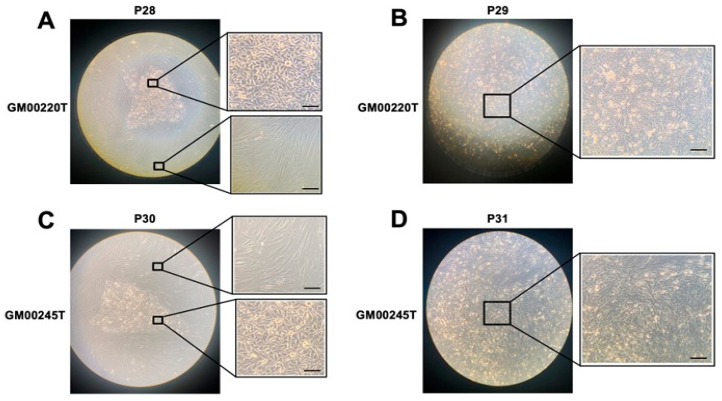
Representative images of the spontaneous transformation of MD cells at culture passages 28–31. Cells routinely cultured in Petri dishes were observed on an inversed microscope cells at the indicated passage. (GM00220T at passage 28 and 29 (**A**,**B**) and GM0024T at passage 30 and 31 (**C**,**D**), respectively. The black bars represent 100 µm.

**Figure 10 biomolecules-13-01746-f010:**
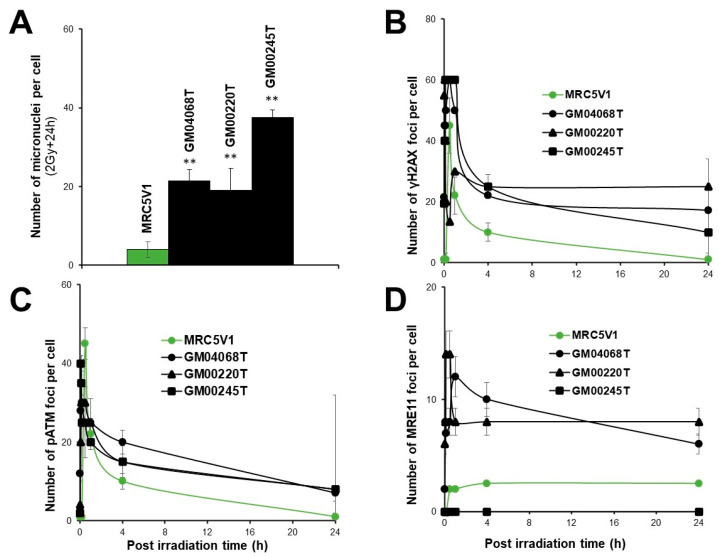
Radiobiological characterization of the spontaneously transformed MD GM04068T, GM0220T, and GM00245T. The MRC5VI cells was used as radioresistant controls. Micronuclei (**A**), γH2AX- (**B**), pATM- (**C**) and MRE11- (**D**) immunofluorescence were applied to cells exposed to two Gy X-rays. For each biomarker, each plot represents the mean ± SEM of three replicates. The inserts show representative images. ** asterisks represent a *p* < 0.0001 difference by comparison to controls data, respectively.

**Figure 11 biomolecules-13-01746-f011:**
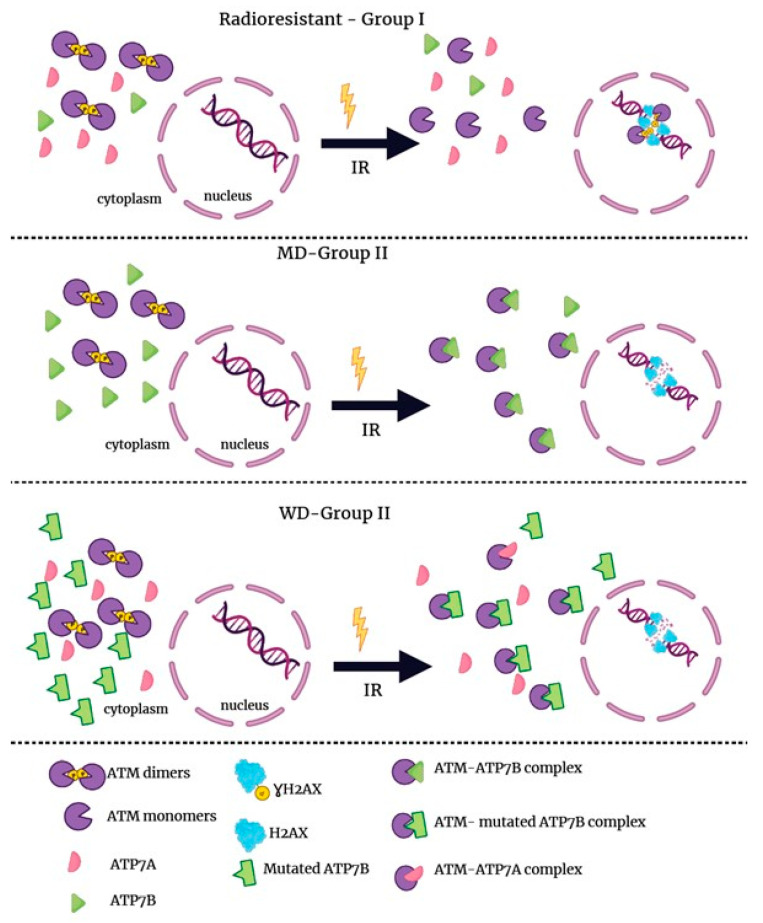
Radiobiological characterization of MD and WD cells. Schematic illustration of the RIANS model. In the case of radioresistant cells, MD, and WD cells, the model shows that radiation-induced ATM monomers diffuse in the nucleus. Once in the nucleus, ATM monomers initiate DSB recognition and repair via the phosphorylation of H2AX histones. In MD cells, ATP7A mutations result in the disappearance or the non-availability of ATP7A Conversely the ATP7B protein is overexpressed in cytoplasm and may sequestrate ATM in the cytoplasm, which may delay the RIANS. In WD cells, ATP7B mutations cause an impaired cytoplasmic localization of the ATP7B protein and its overexpression. However, the ATP7A protein is also expressed. After irradiation, a number of ATM–ATP7B complexes, and less frequently of ATM–ATP7A protein result in a delayed RIANS. IR: ionizing radiation.

**Table 1 biomolecules-13-01746-t001:** Major genetic and clinical features of the untransformed cell lines used in this study.

Cell Lines	Origin	Syndrome	Genetic Features *	Clinical Features *
1BR3	ECACC	-	Apparently healthy	Apparently healthy radioresistant
MRC5	ECACC	-	Apparently healthy	Apparently healthy radioresistant
Hs27	ECACC	-	Apparently healthy	Apparently healthy radioresistant
AT4BI	COPERNIC	AT	*ATM* mutations	Hyper-radiosensitive
GM04068	Coriell Institute	Relative of MD patients	abnormal metallothionein gene regulation in response to Cu	Mother of GM00245
GM01057	Coriell Institute	MD	abnormal metallothionein gene regulation in response to Cu	Fibroblasts exhibit elevated Cu concentration
GM00220	Coriell Institute	MD	abnormal metallothionein gene regulation in response to Cu; culture shows qualitative changes in Mc1 mRNA	Fibroblasts exhibit elevated Cu levels; positive family history; similarly affected brother and cousin
GM00245	Coriell Institute	MD	abnormal metallothionein gene regulation in response to Cu	Fibroblasts exhibit elevated Cu levels
GM05762	Coriell Institute	Relative of MD patients	-	No evidence of WD; rheumatoid arthritis since age 21; mother of the 2 affected daughters GM05339, GM05341
GM05339	Coriell Institute	WD	46,XX,1qh+.arr Xq13.2q21.1(72592523-77910651)x2 hmz,15q11.1q11.2(20601541-21939811)x3	Onset at age 13 with jaundice, evidence of chronic liver disease, presence of Kayser-Fleischer rings, & low ceruloplasmin
GM05341	Coriell Institute	WD	-	Diagnosed as asymptomatic at age 12 with low serum ceruloplasmin, elevated urinary & hepatic Cu levels; no evidence of liver disease or central nervous system dysfunction;

* This information has been provided by the CORIELL institute.

**Table 2 biomolecules-13-01746-t002:** Summary of the major biological and radiobiological features of MD and WD fibroblasts.

	MD Cells	WD Cells
Gene	*ATP7A* mutations	*ATP7B* mutations
Major clinical feature	Abnormally high Cu accumulation in other tissues than liver	Abnormally high Cu accumulation in the liver
Cell survival (SF2)	30–60% (Radioresistance SF2 > 60%)	58–70% (Radioresistance SF2 > 60%)
Micronuclei yield	Not significant Increases with Cu-pretreatment	
Number of γH2AX foci	Lower at 10 min post-irradiation than controls—Delayed RIANS	
Number of pATM foci	Lower at 10 min post-irradiation than controls—Delayed RIANS	
Number of MRE11 foci	Lower than controls. Early MRE11 foci after Cu pre-treatment	Lower than controls. Early and late MRE11 foci after Cu pre-treatment
Gene expression	ATP7A: No/low expression ATP7B: very high expression	ATP7A: Cytoplasmic expression ATP7B: very high expression
Cytoplasmic protein–protein interactions	ATP7B-ATM complexes	ATP7B-ATM complexes Some ATP7A-ATM complexes
Cellular transformation in culture	Spontaneous immortalization from passages 28–31	No signs of immortalization or senescence observed up to passages 28–31

## Data Availability

All the data can be provided on reasonable request.

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
