# Peer review of "Response of Fibroblasts from Menkes’ and Wilson’s Copper Metabolism-Related Disorders to Ionizing Radiation: Influence of the Nucleo-Shuttling of the ATM Protein Kinase"

_biomolecules, 2023, doi:10.3390/biom13121746_

Round 1

Reviewer 1 Report

Comments and Suggestions for Authors

In this research work the use of fibroblasts originating from copper metabolism deficiencies are used. The work in general is original and very interesting since it relates this type of deficiency with response to ionizing radiation. The authors provide logical explanation for their results and discussion. In some points though one can argue on the methodology or interpretation. In addition the authors are asked to provide more details in some cases.

More specifically :

1. The fibroblasts are well-known to have a delayed cell growth after late passages eg. 30 or so. Hiw the authors have dealt with this and performed all these experiments?

The details provided for each cell line should be accompanied by cell growth data.

2. The figure 4 and 5 data with foci kinetics has been done using how many experimental points?

Only one dose has been used, 2 Gy of X rays?

Radiation response and results can be affected by dose. 

3. The smaller or larger foci can be an indication also of DNA damage complexity and not only the interpretation the authors provide. How are they sure about this?

4. The genomic instability detected using micronuclei assay as known is done for high doses 2 Gy and above. Therefore again can argue on the conclusions for radiosensitivity or radioresistance when it comes to genomic instability. The authors need to explain much better if the radioresistance for example found relates to prior genomic instability of the cells due to for example ATM or other protein dysfunction. 

5. The cell lines from the specific syndromes, have been known to have other genomic changes?

6. Last but not least, the authors perform western blotting and always in these cases, there is need to explain and provide not only the original gels which they do and it is very good but also the intensity values changes as analyzed.

Concluding, the amount of data is impressive but regarding the clinical phenotype of the patients, can one use fibroblasts to predict radiation response of these people? 

How the authors work can aid towards this? 

Comments on the Quality of English Language

The English language used is sufficient. Only minor edits and a good proofreading must be done. 

Author Response

Reply to reviewer 1

We thank the reviewer for his/her comments.

In this research work the use of fibroblasts originating from copper metabolism deficiencies are used. The work in general is original and very interesting since it relates this type of deficiency with response to ionizing radiation. The authors provide logical explanation for their results and discussion. In some points though one can argue on the methodology or interpretation. In addition the authors are asked to provide more details in some cases.

More specifically :

The fibroblasts are well-known to have a delayed cell growth after late passages eg. 30 or so. Hiw the authors have dealt with this and performed all these experiments?

You are fully right. As mentioned in materials and methods, to the notable exception of the experiments performed with spontaneously transformed cells, all the experiments has been done with passages lower than 12. During the cell culture, no significant change of shape, features has been noticed in the 1-12 passages cells. See modified text in materials and Methods and section 3.7

The details provided for each cell line should be accompanied by cell growth data.

At the passages 1-12, the average doubling time of all the cell lines used in this study was 28 ± 4 hours, to the notable exception of the ATM-mutated cells (30 ± 1 hours). See modified text in materials and methods.

The figure 4 and 5 data with foci kinetics has been done using how many experimental points?

OK see modified text in captions of figures 4 to 6.

Only one dose has been used, 2 Gy of X rays?. Radiation response and results can be affected by dose.

OK We agree The great majority of radiotherapy protocols involved a dose of 2 Gy per session.  Furthermore, the SF2 has been abundantly shown to be the most useful and representative parameter for radiosensitivity. Besides, in a recent paper, we provided quantified intercorrelation between other radiosensitivity parameters, whether cellular, cytogenetic or molecular ones (ref 36) so that the SF2 values obtained in this paper can be easily compared with those from other syndromes and with some other radiosensitivity biomarkers. See modified text in section 2.2. Finally, to provide a dose-response with so many data time points nd cell lines and conditions was not possible practically.

The smaller or larger foci can be an indication also of DNA damage complexity and not only the interpretation the authors provide. How are they sure about this?

Indirectly yes. As already mentioned, we have encountered this situation in a number of stress conditions : data obtained with H202, a single- and double-strand breaks inducer showed that the phophorylation of gH2AX is always due to the an ATM activation and induction of DNA double-strand breaks (ref 44) : as far as the concentraiton of H202 increases, the number of SSB and therefater of DSB increases : gH2AX foci only appear when DSB are created. However, according the SSB/DSB ratio, additional SSB may contribute to the decondensation of chromatin  and therefore unwinds the strands by letting appearing the individual phosphorylated H2AX histones (ref 44). A dedicated paper is in preparation about this subject. See modified text at the end of section 3.3.

The genomic instability detected using micronuclei assay as known is done for high doses 2 Gy and above. Therefore again can argue on the conclusions for radiosensitivity or radioresistance when it comes to genomic instability. The authors need to explain much better if the radioresistance for example found relates to prior genomic instability of the cells due to for example ATM or other protein dysfunction. 

Literature data clearly shows that the spontaneous levels of DNA or chromosome damage or micronuclei is not correlated to the radiation response and their radiation-induced counterparts. Recent micronucleus data obtained from 200 human fibroblasts and various genetic syndromes associated with genomic instability or not confirm such statement (ref 36). Micronuclei are known to be the consequences of unrepaired DNA strand breaks that are propagated in G2/M phase. This is particular true here since a great majority of cells were irradiated in plateau phase of growth and since micronuclei were observed 24 h post-irradiation. Hence, if a lack of cell cycle control in G1 is a feature of WD/MD cells, such feature will favour the rapid formation of micronuclei. But the impairements of DSB recognition or repair are independent of the impairment of cell cycle checkpoint. In other terms , for the same radiosensitivity, we can have different genomic instability and cell cycle arrest impairments. The reference 49 is dedicated ot this question by making the difference between radiosensitivity and radiosusceptibility. This is may be particularly true after a Cu pre-treatment that contrinbutes to impair cell cycle checkpoint as observed and published elsewhere.

The cell lines from the specific syndromes, have been known to have other genomic changes?

Unfortunately, since our study is one of the first Last but not least, the authors perform western blotting and always in these cases, there is need to explain and provide not only the original gels which they do and it is very good but also the intensity values changes as analyzed.

See modified figure and text in section 3.6

Concluding, the amount of data is impressive but regarding the clinical phenotype of the patients, can one use fibroblasts to predict radiation response of these people?  How the authors work can aid towards this? 

OK You are right See modified text in Discussion about the releance of skin fibroblasts as cellular model.

Reviewer 2 Report

Comments and Suggestions for Authors

El Nachef  and colleagues present a comprehensive radiobiological characterization of cell lines from Menkes’ and Wison’s diseases (MD and WD respectively). The array of methods used in the paper is adequate to portray the effects of ionizing radiation (IR) on the MD and WD cell line models and the data sufficiently support Authors’ conclusions. The research is presented clearly and is of importance to the readership of the special issue and the journal. The manuscript can be considered for publication by the Editors after a careful consideration of the following minor points:

·       The Authors present results from the clonogenic assay at a single dose of 2Gy. While there is enough statistical difference to present survival variations in response to the IR between MD,WD, and control cells, there is a lack of explanation why this particular dose was chosen. Moreover, in the context of WD patients possibly undergoing radiotherapy (especially regarding fractionation), better characterization of radiosensitivity/radioresistance of those cells would be achieve by reporting alpha/beta ratios and cell survival curves for the linear-quadratic model.

·       Does copper (II) sulfate treatment have any influence on long-term clonogenic survival of MD and WD cells? Similarly, dose the pre-treatment with e.g., chelators in case of the  WD cells changes their sensitivity to IR?

·       The overall quality of the figures is compromised. Please consider using lossless export formats for figure preparation and export/import to the manuscript.

·       Same issue with the exemplary fluorescence images – please retain high resolution. Additionally, those insert lack a description what particular time point they represent throughout all figures. 

·       How were the foci pATM or yH2AX quantified (e.g., manually using ImageJ or using automated method)? How the fluorescent positive cells were discriminated from the fluorescent negative cells (classified based on MFI threshold, foci quantification, or “read-by-eye”?) – please provide this information in the method section. 

Comments on the Quality of English Language

Minor English check-up required - for example line 110  (...) focus on the tumor.

Please refrain from using ellipsis when giving examples (e.g., lines 76 and 95).

Author Response

Reply to reviewer 2

We thank the reviewer for his/her comments.

El Nachef  and colleagues present a comprehensive radiobiological characterization of cell lines from Menkes’ and Wison’s diseases (MD and WD respectively). The array of methods used in the paper is adequate to portray the effects of ionizing radiation (IR) on the MD and WD cell line models and the data sufficiently support Authors’ conclusions. The research is presented clearly and is of importance to the readership of the special issue and the journal. The manuscript can be considered for publication by the Editors after a careful consideration of the following minor points:

The Authors present results from the clonogenic assay at a single dose of 2Gy. While there is enough statistical difference to present survival variations in response to the IR between MD,WD, and control cells, there is a lack of explanation why this particular dose was chosen. Moreover, in the context of WD patients possibly undergoing radiotherapy (especially regarding fractionation), better characterization of radiosensitivity/radioresistance of those cells would be achieve by reporting alpha/beta ratios and cell survival curves for the linear-quadratic model.

OK We agree The great majority of radiotherapy protocols involved a dose of 2 Gy per session.  Furthermore, the SF2 has been abundantly shown to be the most useful and representative parameter for radiosensitivity. Besides, in a recent paper, we provided quantified intercorrelation between other radiosensitivity parameters, whether cellular, cytogenetic or molecular ones (ref 36) so that the SF2 values obtained in this paper can be easily compared with those from other syndromes and with some other radiosensitivity biomarkers. See modified text in section 2.2. Finally, to provide a dose-response with so many data time points nd cell lines and conditions was not possible practically.

We have added the alpha beta ratios corresponding to our data : these values are mathematical agreement with a collection of data from more than 200 celll lines (see ref 36 and paper in preparation). See modified text in section 3.1. Alpha/beta ratios will be very high because of the lack of curvature (low beta).

Does copper (II) sulfate treatment have any influence on long-term clonogenic survival of MD and WD cells? Similarly, dose the pre-treatment with e.g., chelators in case of the  WD cells changes their sensitivity to IR?

The data shown here represent an enormous amount of data and we did not apply Cu pretreatment or any other treatment to survival curve since clonogenic assay is heavy and time-consuming. Furthermore, a number of controls and intermediate data should be obtained before analysing the final curves since the following question is raised :should the Cu or chelator -pretreatment be repeated before each dose with the same concentrations, and repeated every day up to the appearance of the colonies? However, even if Cu pre-treatment has not been applied to survival assay, we can easily calculate the SF2 by using mathematical links established between SF2 and other biomarkers of radiosensitivity (ref 36).

The overall quality of the figures is compromised. Please consider using lossless export formats for figure préparation and export/import to the manuscript.

OK see all the modified figuresSame issue with the exemplary fluorescence images – please retain high resolution. Additionally, those insert lack a description what particular time point they represent throughout all figures. 

 OK see all the modified figures and captions

  • How were the foci pATM or yH2AX quantified (e.g., manually using ImageJ or using automated method)? How the fluorescent positive cells were discriminated from the fluorescent negative cells (classified based on MFI threshold, foci quantification, or “read-by-eye”?) – please provide this information in the method section. 

OK see modified text in Materials and methods

Comments on the Quality of English Language

Minor English check-up required - for example line 110  (...) focus on the tumor.

OK see modified text

Please refrain from using ellipsis when giving examples (e.g., lines 76 and 95).

OK see modified text

Round 2

Reviewer 1 Report

Comments and Suggestions for Authors

The authors have responded in all my comments but the use of one cell line and one dose is a disadvantage of their work. It remains in the policy of the Journal if it can be accepted or not.